# Semi-Synthesis of *N*-Aryl Amide Analogs of Piperine from *Piper nigrum* and Evaluation of Their Antitrypanosomal, Antimalarial, and Anti-SARS-CoV-2 Main Protease Activities

**DOI:** 10.3390/molecules27092841

**Published:** 2022-04-29

**Authors:** Rattanaporn Wansri, Aye Chan Khine Lin, Jutharat Pengon, Sumalee Kamchonwongpaisan, Nitipol Srimongkolpithak, Roonglawan Rattanajak, Patcharin Wilasluck, Peerapon Deetanya, Kittikhun Wangkanont, Kowit Hengphasatporn, Yasuteru Shigeta, Jatupol Liangsakul, Aphinya Suroengrit, Siwaporn Boonyasuppayakorn, Taksina Chuanasa, Wanchai De-eknamkul, Supot Hannongbua, Thanyada Rungrotmongkol, Supakarn Chamni

**Affiliations:** 1Pharmaceutical Sciences and Technology Program, Faculty of Pharmaceutical Sciences, Chulalongkorn University, Bangkok 10330, Thailand; 6176127433@student.chula.ac.th (R.W.); 6373015133@student.chula.ac.th (A.C.K.L.); 2Department of Pharmacognosy and Pharmaceutical Botany, Faculty of Pharmaceutical Sciences, Chulalongkorn University, Bangkok 10330, Thailand; taksina.c@pharm.chula.ac.th (T.C.); wanchai.d@chula.ac.th (W.D.-e.); 3Natural Products and Nanoparticles Research Unit (NP2), Chulalongkorn University, Bangkok 10330, Thailand; 4National Center for Genetic Engineering and Biotechnology (BIOTEC), National Science and Technology Development Agency, Thailand Science Park, Pathum Thani 12120, Thailand; jutharat.pen@biotec.or.th (J.P.); sumaleek@biotec.or.th (S.K.); nitipol.sri@biotec.or.th (N.S.); roonglawan@biotec.or.th (R.R.); 5Center of Excellence for Molecular Biology and Genomics of Shrimp, Department of Biochemistry, Faculty of Science, Chulalongkorn University, Bangkok 10330, Thailand; patcharinwilasluck@gmail.com (P.W.); peerapol302@gmail.com (P.D.); kittikhun.w@chula.ac.th (K.W.); 6Molecular Crop Research Unit, Department of Biochemistry, Faculty of Science, Chulalongkorn University, Bangkok 10330, Thailand; 7Center for Computational Sciences, University of Tsukuba, 1-1-1 Tennodai, Tsukuba 305-8577, Ibaraki, Japan; heng.kowit@gmail.com (K.H.); shigeta@ccs.tsukuba.ac.jp (Y.S.); 8Scientific and Technological Research Equipment Centre, Chulalongkorn University, Bangkok 10330, Thailand; jatu.p@chula.ac.th; 9Applied Medical Virology Research Unit, Department of Microbiology, Faculty of Medicine, Chulalongkorn University, Bangkok 10330, Thailand; aphinya.su@chula.md (A.S.); siwaporn.b@chula.ac.th (S.B.); 10Natural Product Biotechnology Research Unit, Chulalongkorn University, Bangkok 10330, Thailand; 11Center of Excellence in Computational Chemistry (CECC), Department of Chemistry, Faculty of Science, Chulalongkorn University, Bangkok 10330, Thailand; supot.h@chula.ac.th; 12Center of Excellence in Biocatalyst and Sustainable Biotechnology, Department of Biochemistry, Faculty of Science, Chulalongkorn University, Bangkok 10330, Thailand; thanyada.r@chula.ac.th; 13Program in Bioinformatics and Computational Biology, Graduate School, Chulalongkorn University, Bangkok 10330, Thailand

**Keywords:** black pepper, piperine analogs, semi-synthesis, antimalaria, antitrypanosoma, anti-SARS-CoV-2 main protease, molecular dynamic

## Abstract

*Piper nigrum*, or black pepper, produces piperine, an alkaloid that has diverse pharmacological activities. In this study, *N*-aryl amide piperine analogs were prepared by semi-synthesis involving the saponification of piperine (**1**) to yield piperic acid (**2**) followed by esterification to obtain compounds **3**, **4**, and **5**. The compounds were examined for their antitrypanosomal, antimalarial, and anti-SARS-CoV-2 main protease activities. The new 2,5-dimethoxy-substituted phenyl piperamide **5** exhibited the most robust biological activities with no cytotoxicity against mammalian cell lines, Vero and Vero E6, as compared to the other compounds in this series. Its half-maximal inhibitory concentration (IC_50_) for antitrypanosomal activity against *Trypanosoma brucei rhodesiense* was 15.46 ± 3.09 μM, and its antimalarial activity against the 3D7 strain of *Plasmodium falciparum* was 24.55 ± 1.91 μM, which were fourfold and fivefold more potent, respectively, than the activities of piperine. Interestingly, compound **5** inhibited the activity of 3C-like main protease (3CL^Pro^) toward anti-SARS-CoV-2 activity at the IC_50_ of 106.9 ± 1.2 μM, which was threefold more potent than the activity of rutin. Docking and molecular dynamic simulation indicated that the potential binding of **5** in the 3CL^pro^ active site had the improved binding interaction and stability. Therefore, new aryl amide analogs of piperine **5** should be investigated further as a promising anti-infective agent against human African trypanosomiasis, malaria, and COVID-19.

## 1. Introduction

During the 20th century, the world’s population faced several life-threatening infectious diseases. The 1920 trypanosomiasis epidemic lead to the death of millions of people in African countries [1,2,3]. The human African trypanosomiasis is caused by two parasites, which are *Trypanosoma brucei gambiense* and *Trypanosoma brucei rhodesiense*, transmitted by tsetse flies (*Glossina* sp.). Trypanosomiasis has been listed as one of the four major neglected tropical diseases, and the World Health Organization has initiated the goal of eliminating its transmission by 2030 [4,5]. Malaria, currently endemic across broad areas, including America, Asia, and Africa, is caused by *Plasmodium* parasites, including *P. falciparum*, *P. vivax*, *P. ovale*, *P. malariae*, and *P. knowlesi* [6,7]. The parasites are transmitted to people through the bites of infected female *Anopheles* mosquitoes. *P. falciparum* has been reported as a major risk for human death. In 2019, an estimated 200 million malaria cases were reported globally, nearly half a million people died of malaria, and more than 60% of patients were children [8]. The resistance of malaria parasites to the general antimalarial drugs such as chloroquine, mefloquine, pyrimethamine, and artemisinin is of concern [9,10,11]. Thus, the discovery and development of alternative antimalarial agents are highlighted as the continued research.

Since December 2019, the newly emerged SARS-CoV-2 has critically infected more than 200 million people and drastically caused over 4 million deaths globally [12]. An epidemic of the infectious coronavirus disease or COVID-19 has continued to rapidly spread worldwide. Therefore, safe and effective vaccines and promising anti-SARS-CoV-2 agents are urgently needed to prevent and treat COVID-19 illness [13,14]. The 3-chymotrypsin-like main protease (3CL^Pro^) of SARS-CoV-2 has been reported as an essential enzyme for viral replication and transcription [15]. Therefore, 3CL^Pro^ has been used as a potential drug target, and a screening assay has been developed for the identification of anti-SARS-CoV-2 drug leads [16,17].

Natural products are well recognized as promising sources of biologically active compounds due to their structural complexity and diversity. In addition, herbal medicines have a long history of use in traditional medicine and their specific efficacy and safety are well documented. A notable example is the use of sweet wormwood *Artemisia annua* containing artemisinin in Chinese traditional medicine for the treatment of malaria [18]. The discovery and development of potential therapeutic agents as drug leads or medicines against the existing and remerging infectious diseases have been intensively explored. In response to the COVID-19 pandemic, the Thai medicinal herbs *Andrographis paniculata* and *Boesenbergia rotunda*, containing the active phytochemicals andrographolide and panduratin A, respectively, have been investigated and identified as promising anti-SARS-CoV-2 agents, along with several other local herbs [19,20]. 

The dried seed of the *Piper nigrum* plant, known as black pepper, produces an alkaloid named piperine (**1**), which regulates its pungent smell and spicy taste. Piperine exhibits diverse pharmacological activities, including antibacterial, antifungal, bioavailability enhancing, and non-genotoxic properties [21,22,23]. Its antiviral activity against coxsackie virus type B3 and human rhinovirus type 2 (HRV2) has also been reported [24]. In addition, various piperine analogs demonstrate interesting biological activities such as antimicrobial, anticancer, and antituberculosis [25]. 

We have prepared piperine and semi-synthetic *N*-aryl amide analogs and evaluated their cytotoxicity, antitrypanosomal, antimalarial, and anti-SARS-CoV-2 main protease 3CL^Pro^ activities. Furthermore, the binding interaction of the active piperine analogs with 3CL^Pro^ were demonstrated by docking and molecular dynamic (MD) simulation. The findings of this study support the development of potential anti-infective agents with socio-economic advantages for the treatment of human African trypanosomiasis, malaria, and COVID-19. 

## 2. Results and Discussion 

### 2.1. Extraction of Piperine and Semi-Synthesis of N-Aryl Amide Analogs of Piperine

Piperine (**1**) was isolated from the dried black pepper seed (*Piper nigrum*) powder on a modified extraction procedure [26,27]. Pure natural piperine was afforded at 6.43% yield based on the dry weight of the black pepper seed powder. Next, the saponification of piperine under basic conditions produced piperic acid (**2**) at an acceptable yield (18.03%). The *N*-aryl amide analogs of piperine **3**–**5** were designed by the replacement of the piperidine moiety with aryl amine-containing phenyl, 2,4-dimethoxyphenyl, and 2,5-dimethoxyphenyl substituents to improve its drug-likeness properties. Regarding the prediction of absorption, distribution, metabolism, and excretion (ADME) parameters, *N*-aryl amide analogs of piperine **4** and **5** displayed improved H-bond acceptors, H-bond donors, and lipophilicity that exceeded the parent piperine while maintaining its the bioavailability score (see Appendix A) [28,29,30].

The semi-syntheses of *N*-aryl piperamide analogs **3****–5** were performed using piperic acid and amine reagents, including aniline, 2,4-dimethoxyaniline, and 2,5-dimethoxyaniline as the starting materials for amidation (Figure 1). Several amide synthetic conditions were employed, including the conversion of the acid motif to an activated acyl chloride by thionyl chloride or oxalyl chloride and the formation of highly reactive *O*-acylisourea derived from amide coupling reagents, such as dicyclohexylcarbodiimide and 1-ethyl-3-(3-dimethylaminopropyl)carbodiimide (EDCI) in the presence of 4-dimethylaminopyridine [31,32,33,34]. However, these conditions did not deliver the desired *N*-aryl piperamides **4** and **5** possibly due to the resonance effects of both (3,4-methylenedioxyphenyl)-penta-2,4-dienoyl moiety of piperic acid and dimethoxy substituted phenyl groups of amines, which resulted in the low reactivities of both electrophiles and nucleophiles, respectively. Combining EDCI and 1-hydroxy-benzotriazole (HOBt), the aryl amide analogs of piperine were obtained in suitable yields (42–47%). Herein, we report for the first time, the chemical and biological aspects of the methoxy-substituted phenyl piperamides **4** and **5**. 

Compounds **1****–5** were structurally characterized by spectroscopic techniques. Piperine (**1**) demonstrated characteristic piperidine ^1^H and ^13^C chemical shifts at 1.53, 1.64, and 3.57 ppm and at 25.5, 26.6, 27.6, 43.5, and 46.6 ppm, respectively, along with ^13^C chemical shift of amide at 165.3 ppm. Piperic acid (**2**) has a characteristic ^13^C chemical shift of the carboxylic acid motif at 167.9 ppm. Compound **3** displayed the ^1^H and ^13^C chemical shifts of the additional phenyl moiety at 7.30 and 7.74 ppm and 140.7, 129.6, 124.2, and 120.2 ppm. Compound **4** showed the characteristic peaks of 2,4-dimethoxy substituents at 3.78 and 3.86 ppm for ^1^H and 55.8 and 56.3 ppm for ^13^C chemical shifts. The proton coupling constant between H-3′ and H-5′ was 2.8 Hz, which referred to aromatic *meta* proton coupling. The proton coupling constant between H-5′ and H-6′ was 8.8 Hz, which referred to aromatic *ortho* proton coupling. Both coupling constants corresponded to the structure of compound **4**. Compound **5** showed ^1^H chemical shifts at 3.74 and 3.83 ppm and ^13^C chemical shifts at 55.9 and 56.8 ppm corresponding to the 2,5-dimethoxy substituents. The coupling constants also correlated to the structure of compound **5**, including aromatic *ortho* proton coupling between H-3′ and H-4′ at 8.8 Hz and aromatic *meta* proton coupling between H-4′ and H-6′ at 2.8 Hz. In addition, the chemical structures of all the compounds were confirmed by the high-resolution mass analysis (see Appendix A for the spectra). The resulting compounds **1****–5** were further evaluated for their antitrypanosomal, antimalarial, and anti-SARS-CoV-2 main protease 3CL^Pro^ activities.

### 2.2. Cytotoxicity against Vero and Vero E6 Cell Lines

Next, compound **1****–5** were examined their cytotoxicity against kidney epithelial Vero and infected Vero E6 cell lines. As the results shown in Table 1, piperine (**1**), piperic acid (**2**), and analogs **3**–**5** were non-toxic against Vero cells at the concentration of 100 μM. In addition, piperine (**1**) demonstrated cytotoxicity at the half maximal effective concentration (EC_50_) of 131.67 ± 2.91 μM, while piperic acid (**2**) and analogs **3**–**5** were non-toxic to Vero E6 cell lines at the concentration of 500 μM.

### 2.3. In Vitro Evaluation of Antitrypanosomal, Antimalarial, and Anti-SARS-CoV-2 Main Protease (3CL^Pro^) Activities

The biological activities of compounds **1****–****5** were investigated by focusing on African trypanosomiasis, malaria, and COVID-19. Antitrypanosomal activity was evaluated against *T. brucei rhodesiense* (Figure 2a), and antimalarial activity was evaluated against 3D7 *P. falciparum*, a wild-type drug-sensitive strain (Figure 2b). Piperine (**1**), piperic (**2**), and analogs (**3****–5**) demonstrated concentration-dependent inhibitory activity against both parasites (Table 2).

Piperine (**1**) and amide **3** possessed similar mild antitrypanosomal inhibitory potency, while compounds **4** and **5** demonstrated approximately threefold and fourfold more potency than the mother compound **1**. Piperic acid (**2**) was inactive against *T*. *brucei rhodesiense* having no inhibitory effect at 100 μM. The *N*-aryl amide analogs (**3**–**5**) exhibited approximately twofold to threefold improved antimalarial inhibition compared to the naturally derived compounds **1** and **2**. 

These results emphasized the importance of the methoxy-substituted phenyl amide scaffold of the piperine analog **4** and **5**, which is related to the improvement of lipophilicity, a higher number of oxygen atoms, and a higher number of solvated hydrogen bond donors and acceptors [35]. The results indicated that compounds **4** and **5**, which contained 2,4 and 2,5-dimethoxy phenyl amide, respectively, had the most interesting antitrypanosomal and antimalaria activities in this series. However, they demonstrated mild inhibition compared to pentamidine and pyrimethamine, which were used as the positive controls for antitrypanosomal and antimalarial activities, respectively. Nonetheless, these compounds may serve as targets for the development of antitrypanosomals and antimalarials.

### 2.4. In Vitro Evaluation of Anti-SARS-CoV-2 3C-like Main Protease Activity

The anti-SARS-CoV-2 3C-like main protease activities of compounds **1****–5** were evaluated in a 3CL^Pro^ inhibitory assay employing E(EDANS)TSAVLQSGFRK(DABCYL) as the fluorogenic substrate compared with rutin (100 μM) as the positive control (Figure 3) [36]. Compounds **1**–**5** were initially screened at a concentration of 100 μM. The results demonstrated that piperine (**1**) and 2,5-dimethoxy phenyl amide of piperine (**5**) had twofold more potent inhibitory activity against 3CL^Pro^ than that of rutin. At 100 μM, both **1** and **5** exhibited approximately 70% inhibition toward 3CL^Pro^. Next, IC_50_ of piperine (**1**) and analog **5** against 3CL^Pro^ inhibition were further obtained at 178.4 ± 1.2 μM and 106.9 ± 1.2 μM, respectively. Analog **5** improved anti-SARS-CoV-2 activity better than piperine (**1**), with threefold more potency than rutin (Table 3).

### 2.5. Predicted Binding Mode of Piperine (**1**), Piperic Acid (**2**), and N-Aryl Amide Analogs of Piperine (**3**–**5**) toward SARS-CoV-2 3C-like Main Protease 

A molecular docking study was performed to reveal the binding modes of piperine (**1**) and its derivatives (**2**–**5**) toward the inhibition of 3CL^Pro^. The docking simulation illustrated that the compounds in this series have two different binding conformations at the 3CL^Pro^ active site with the insertion of (i) the piperidine ring (compounds **1** and **3**) or (ii) the 1,3-benzodioxole group (compounds **2**, **4**, and **5**) into the S2 pocket (Figure 4a). To evaluate the ligand-binding stability, these complexes were performed by the molecular dynamics (MD) simulations for 100 ns. In Figure 4b, the plot of the distance between the center of the mass of each ligand and the catalytic dyad (H41 and C145) shows that compounds **1** and **5** could bind to the 3CL^Pro^ catalytic site along 100 ns relative to rutin, the known anti-SARS-CoV-2 3CL^Pro^ agent [36,37,38,39,40,41]. In contrast, the other compounds could not maintain their positions in the binding pocket. The MD simulations of the piperine (**1**) and the 2,5-dimethoxy substituted phenyl piperamide **5** that demonstrated good inhibitory activity as reported above, were then extended to 200 ns. 

RMSD analysis (Appendix A) indicated that the ligand-bound proteins of the two complexes were relatively stable, as had been demonstrated by the simulation. At the same time, compound **5** revealed remarkably steadier dynamics than the other compounds in this series, in particular after 100 ns (1.59 ± 0.13 Å and 1.24 ± 0.30 Å for compounds **5** and **1**). A series of 5000 snapshots extracted from the last 50 ns was used to estimate the per-residue decomposition binding free energy ($\Delta G_{\mathrm{bind}}^{\mathrm{residue}}$) using MM/GBSA calculations (Figure 4(c1)). The interaction profile in Figure 4(c2) depicted that the binding of piperine **1** and 2,5-dimethoxy substituted phenyl piperamide **5** in the 3CL^Pro^ active site differed and they were stabilized by the 6 and 10 surrounding amino acid residues, respectively, mainly through van der Waals (vdW) interactions. One of the oxygens on the 1,3-benzodioxole group of the parent piperine (**1**) formed a hydrogen bond with the imidazole moiety of H163. In contrast, the 1,3-benzodioxole group of compound **5** interacted with the sulfur atom of M49 through the sulfur–π interaction, which was found to help the ligand tightly packed to the pocket [42,43,44]. Additionally, one of the methoxy groups on the 2,5-dimethoxy phenyl amide motif was strongly stabilized by the residues in the oxyanion region (F140, L141, N142, S144, and C145 with $\Delta G_{\mathrm{bind}}^{\mathrm{residue}}$ in a range from −0.55 to −1.38 kcal/mol, Figure 4(c3)). Two hydrogen bond formations with the E166 backbone and N142 sidechain were important for strengthening the binding of **5**, the modified piperine analog. 

## 3. Materials and Methods

### 3.1. General Experimental Procedures

Commercial reagents were purchased from Tokyo Chemical Industry (TCI). Solvents and reagents were used as received unless otherwise stated. All the solvents such as methanol, ethyl acetate, dichloromethane, and hexane were distilled before use. Commercial anhydrous solvents were dried over 4 Å molecular sieves prior to use. Reactions were carried out in an oven-dried glassware and magnetically stirred with a nitrogen (N_2_) gas balloon for an inert atmosphere. All the reactions were monitored by thin-layer chromatography (TLC) using aluminum silica gel 60F254 (Merck). Bands were identified by UV activity at 256 nm. Yields refer to chromatographically and spectroscopically pure compounds unless otherwise stated. Flash column chromatography was performed using 60 Å silica gel (230–400 mesh) as a stationary phase. Infrared (IR) spectra were measured on a Perkin Frontier Fourier Transform Infrared Spectrometer. Furthermore, ^1^H and ^13^C nuclear magnetic resonance (NMR) spectra were obtained on a Bruker ADVANCE NEO 400 MHz NMR spectrometer. Deuterated acetone-d_6_ ((CD_3_)_2_CO) were served as the internal standard for both ^1^H (2.05 ppm) and ^13^C (29.92 ppm) NMR spectra. Accurate mass spectra were obtained with an Agilent 6540 UHD Q-TOF LC/MS spectrometer.

The in vitro assays were carried out at 37 °C under the suitable concentration of carbon dioxide (CO_2_) gas atmosphere. Minimum Essential Media (MEM) and Earle’s Balanced Salt Solution (EBSS) were received from Hyclone Laboratories Inc., South Logan, UT, USA. Roswell Park Memorial Institute Medium (RPMI) 1640 culture media was obtained from Life Technologies Limited, Paisley, UK. Fetal bovine serum (FBS), L-glutamine, penicillin/streptomycin solution, bathocuproinedisulfonic acid disodium salt, L-cysteine, hypoxanthine, thymidine, sodium pyruvate, and 2-mercaptoethanol were obtained from Gibco, Gaithersburg, MA, USA. Dimethyl sulfoxide (DMSO) was purchased from Merck Millipore, Billerica, MA, USA. 4-(2-hydroxyethyl)-1-piperazineethane sulfonic acid (HEPES), gentamicin, human serum, and resazurin were purchased from Sigma-Aldrich. Albumax I was purchased from Life Technologies, Grand Island, NY, USA. Peptide E(EDANS)TSAVLQSGFRK(DABCYL) was obtained from Biomatik, Kitchener, ON, Canada. Dithiothreitol (DTT) was received from GoldBio, St Louis, MO, USA. 

### 3.2. Extraction and Isolation of Piperine from Dried Piper nigrum Seed 

Dried black pepper seed (*Piper nigrum*) was purchased from the local retail market in Bangkok, Thailand. The dried seed was grounded and packed in a muslin bag. The black pepper powder (1 kg) was macerated in 95% *v*/*v* ethanol (3 L) for 3 days at room temperature. The ethanolic extract was filtered and evaporated under reduced pressure to obtain the crude ethanolic extract. The maceration was repeated two more times. The resulting crude extracts were combined and dissolved with 95% *v*/*v* ethanol (200 mL). An aqueous solution of 10% potassium hydroxide (KOH) was added (200 mL) for deresinification. The mixture was stirred for 5 min and kept at room temperature overnight to allow the precipitation of piperine (1). Then, the mixture was filtered and the resulting solid of piperine was washed with water (100 mL, 3 times) to remove the water-soluble residues. The resulting solid was recrystallized in dichloromethane, which afforded 64.25 g of yellow crystalline piperine. This modified the extraction method [26,27], And yielded 6.43% piperine based on the dry weight of the dried *Piper nigrum* seed powder. IR (ATR, ν_max_) 3009, 2940, 2849, 1611, 1633, 1581, 1490, 1433, 1365, 1250, 1193, 1133, 1017, 995, 928, 846, 830, 804, 718, 701, 607, 568 cm^−1^; ^1^H-NMR (400 MHz, acetone-d_6_) δ_H_: 7.31 (dd, *J* = 10.4 and 14.4 Hz, 1H, H-3), 7.11 (d, *J* = 1.6 Hz, 1H, H-12), 6.99 (dd, *J* = 1.6 and 8.0 Hz, 1H, H-7), 6.98 (d, *J* = 8.0 Hz, 1H, H-8), 6.93 (d, *J* = 10.8 Hz, 1H, H-4), 6.85 (d, *J* = 8.0 Hz, 1H, H-5), 6.65 (d, *J* = 14.4 Hz, 1H, H-2), 6.02 (s, 2H, H-10), 3.57 (t, *J* = 5.4 Hz, 4H, H-2′ and H-6′), 1.64 (m, 2H, H-4′), 1.53 (m, 4H, H-3′ and H-5′) ppm; ^13^C-NMR (100 MHz, acetone-d_6_) δ_C_: 165.3 (C-1), 149.3 (C-11), 149.1 (C-9), 142.6 (C-3), 138.9 (C-5), 131.7 (C-6), 126.7 (C-4), 123.4 (C-7), 121.9 (C-2), 109.3 (C-8), 106.4 (C-12), 102.3 (C-10), 46.4 (C-2′), 43.5 (C-6′), 27.6 (C-3′), 26.6 (C-5′), 25.5 (C-4′) ppm; HRMS *m*/*z* 286.1436 ([M + H]^+^, calculated for C_17_H_20_NO_3_, 286.1443). 

### 3.3. Semi-Synthesis of Piperic Acid and N-Aryl Amide Derivatives of Piperine 

#### 3.3.1. Preparation of Piperic Acid (**2**)

Natural piperine (**1**), the starting material, was added to a round-bottom flask (10.0 g, 35.05 mmol) and dissolved in methanol (200 mL), and a 20% *w*/*v* solution of sodium hydroxide (200 mL) was added. The mixture was refluxed for 24 h. After cooling to room temperature, the mixture was filtered to remove the solid residues and concentrated under reduced pressure to remove the volatile solvent. The filtrate was neutralized with 1 M hydrochloric acid (HCl) until it reached pH 1. The resulting yellow solid was collected by filtration and washed with cold water (100 mL, 3 times). The filtrate was further extracted by dichloromethane (100 mL, 3 times), washed with water (100 mL, 3 times), and concentrated under reduced pressure to obtain a yellow solid. The yellow solid was combined and recrystallized in methanol to afford piperic acid (2) at 1.38 g (18.03% yield). IR (ATR, ν_max_) 2916, 1668, 1616, 1596, 1500, 1489, 1446, 1417, 1367, 1309, 1255, 1191, 1147, 1100, 1034, 991, 938, 920, 852, 806, 789, 696, 607, 607, 582 cm^−1^; ^1^H-NMR (400 MHz, acetone-d_6_) δ_H_: 7.40 (m, 1H, H-3), 7.17 (d, *J* = 1.6 Hz, 1H, H-12), 7.03 (dd, *J* = 1.6 and 8.0 Hz, 1H, H-7), 6.98 (d, *J* = 10.0 Hz, 1H, H-4), 6.97 (d, *J* = 5.6 Hz, 1H, H-5), 6.86 (d, *J* = 8.0 Hz, 1H, H-8), 6.04 (s, 2H, H-10), 6.00 (d, *J* = 15.2 Hz, 1H, H-2) ppm; ^13^C-NMR (100 MHz, acetone-d_6_) δ_C_: 167.9 (C-1), 149.6 (C-11), 149.5 (C-9), 146.1 (C-3), 141.0 (C-5), 131.9 (C-6), 125.7 (C-4), 124.0 (C-7), 121.4 (C-2), 109.3 (C-8), 106.6 (C-12), 102.5 (C-10) ppm; HRMS *m*/*z* 219.0650 ([M + H]+, calculated for C_12_H_11_O_4_, 219.0657).

#### 3.3.2. Preparation of N-Aryl Amide Derivative of Piperine (**3**–**5**)

Piperic acid (2) (0.2291 mmol, 50.0 mg) was added to an oven-dried round-bottomed flask and dissolved in dry tetrahydrofuran (5 mL). The mixture was cooled to 0 °C in an ice bath. Next, 1-ethyl-3-(3-dimethylaminopropyl) carbodiimide (0.2291 mmol, 35.57 mg), and HOBt (0.2291 mmol, 30.96 mg) were added. The reaction was stirred at 0 °C for 30 min. Then, the corresponding amine (0.1146 mmol) was added. The reaction was slowly warmed to room temperature, stirred for 16 h, and monitored by TLC using a mixture of ethyl acetate and hexane solution (3:7 *v*/*v*) as the mobile phase. Upon completion, the volatile solvent was removed under reduced pressure; redissolved in ethyl acetate (20 mL); and washed with 1 M HCl (20 mL), saturated sodium bicarbonate (NaHCO_3_) solution (20 mL), and distilled water (20 mL) three times. The organic layer was dried over anhydrous sodium sulfate, filtered, and then evaporated under reduced pressure to afford a crude product. The crude product was purified by flash column chromatography using silica gel as the stationary phase. A mixture of ethyl acetate dichloromethane and hexane solution was used as the eluent. The *N*-aryl amide derivatives of piperine were obtained as a solid. The chemical structures of the resulting derivatives **3**–**5** were characterized by spectroscopic techniques (see Appendix A for the spectra).

(2E,4E)-5-(benzo[d][1,3]dioxol-5-yl)-*N*-phenylpenta-2,4-dienamide (**3**); white solid, 31.64 mg, 47.0%; IR (ATR, ν_max_) 3359, 3194, 2922, 2853, 1651, 1632, 1615, 1534, 1500, 1488, 1468.58, 1441.83, 1368.92, 1338.41, 1037.31, 968.00, 930.16, 745.92, 690.15 cm^−1^; ^1^H-NMR (400 MHz, acetone-d_6_) δ_H_: 9.30 (br s, 1H, NH), 7.74 (dd, *J* = 1.2, 8.8 Hz, 2H, H-2′ and H-6′), 7.43 (ddd, *J* = 3.6, 7.2, 15.2 Hz, 1H, H-3), 7.30 (dd, *J* = 7.6 and 8.8 Hz, 2H, H-3′ and H-5′), 7.15 (d, *J* = 1.6 Hz, 1H, H-12), 7.05 (m, 1H, H-4′), 7.02 (dd, *J* = 1.6 and 8.0 Hz, 1H, H-7), 6.94 (d, *J* = 6.8 Hz, 1H, H-4), 6.93 (d, *J* = 3.6 Hz, 1H, H-5), 6.85 (d, *J* = 8.0 Hz, 1H, H-8), 6.31 (d, *J* = 14.8 Hz, 1H, H-2), 6.03 (s, 2H, H-10) ppm; ^13^C-NMR (100 MHz, acetone-d_6_) δ_C_: 164.9 (C-1), 149.4 (C-11), 149.4 (C-9), 142.3 (C-3), 140.7 (C-1′), 139.8 (C-5), 132.1 (C-6), 129.6 (C-3′), 129.6 (C-5′), 126.0 (C-4), 125.3 (C-2), 124.2 (C-4′), 123.7 (C-7), 120.2 (C-2′), 120.2 (C-6′), 109.3 (C-8), 106.5 (C-12), 102.5 (C-10) ppm; HRMS m/z 294.1121 ([M + H]+, calculated for C_18_H_16_NO_3_, 294.1130).

(2E,4E)-5-(benzo[d][1,3]dioxol-5-yl)-*N*-(2,4-dimethoxyphenyl)penta-2,4-dienamide (**4**); brown solid, 34.30 mg, 42.2%; IR (ATR, ν_max_) 3359, 3193, 3004, 2922, 2851, 1658, 1632, 1526, 1468, 1411, 1253, 1186, 1037, 721 cm^−1^; ^1^H-NMR (400 MHz, acetone-d_6_) δ_H_: 8.46 (br s, 1H, NH), 8.30 (d, *J* = 8.8 Hz, 1H, H-6′), 7.40 (ddd, *J* = 4.6, 6.0 and 14.8 Hz 1H, H-3), 7.14 (d, *J* = 2.0 Hz, 1H, H-12), 7.02 (dd, *J* = 1.6 and 8.0 Hz, 1H, H-7), 6.93 (d, *J* = 6.0 Hz, 1H, H-4), 6.92 (d, *J* = 4.4 Hz, 1H, H-5), 6.85 (d, *J* = 8.0 Hz, 1H, H-8), 6.59 (d, *J* = 2.8 Hz, 1H, H-3′), 6.50 (dd, *J* = 2.8 and 8.8 Hz, 1H, H-5′), 6.46 (d, *J* = 14.8 Hz, 1H, H-2), 6.03 (s, 2H, H-10), 3.86 (s, 3H, 2′-OMe), 3.78 (s, 3H, 4′-OMe) ppm; ^13^C-NMR (100 MHz, acetone-d_6_) δ_C_: 164.4 (C-1), 157.6 (C-4′), 151.0 (C-2′), 149.3 (C-11), 149.3 (C-9), 142.8 (C-1′), 141.7 (C-3), 139.4 (C-5), 132.2 (C-6), 126.1 (C-4), 125.8 (C-2), 123.6 (C-7), 122.2 (C-6′), 109.3 (C-8), 106.5 (C-12), 104.8 (C-5′), 102.5 (C-10), 99.4 (C-3′), 56.3 (2′-OMe), 55.8 (4′-OMe) ppm; HRMS m/z 354.1340 ([M + H]+, calculated for C_20_H_20_NO_5_, 354.1342).

(2E,4E)-5-(benzo[d][1,3]dioxol-5-yl)-*N*-(2,5-dimethoxyphenyl)penta-2,4-dienamide (**5**); brown solid, 38.36 mg, 47.2%; IR (ATR, ν_max_) 3359, 3190, 2922, 2851, 1659, 1632, 1527, 1468, 1411, 1237, 1136, 1039, 721 cm^−1^; ^1^H-NMR (400 MHz, acetone-d_6_) δ_H_: 8.62 (br s, 1H, NH), 8.23 (d, *J* = 2.8 Hz, 1H, H-6′), 7.42 (ddd, *J* = 2.4, 8.0 and 14.8 Hz, 1H, H-3), 7.15 (d, *J* = 1.2 Hz, 1H, H-12), 7.02 (dd, *J* = 1.4 and 8.2 Hz, 1H, H-7), 6.95 (d, *J* = 7.2 Hz, 1H, H-4), 6.94 (d, *J* = 8.8 Hz, 1H, H-5), 6.92 (d, *J* = 8.0 Hz, 1H, H-4′), 6.85 (d, *J* = 8.0 Hz, 1H, H-8), 6.58 (dd, *J* = 2.8 and 8.8 Hz, 1H, H-3′), 6.49 (d, *J* = 14.8 Hz, 1H, H-2), 6.04 (s, 2H, H-10), 3.83 (s, 3H, 2′-OMe), 3.74 (s, 3H, 4′-OMe) ppm; ^13^C-NMR (100 MHz, acetone-d_6_) δ_C_: 164.8 (C-1), 154.8 (C-5′), 149.4 (C-11), 149.4 (C-9), 143.8 (C-1′), 142.4 (C-3), 139.9 (C-4), 132.1 (C-6), 130.2 (C-2′), 126.0 (C-5), 125.5 (C-2), 123.7 (C-7), 112.0 (C-4′), 109.3 (C-8), 108.3 (C-3′), 107.9 (C-6′), 106.5 (C-12), 102.5 (C-10), 56.8 (2′-OMe), 55.9 (5′-OMe) ppm; HRMS m/z 354.1339 [M + H]+, calculated for C_20_H_20_NO_5_, 354.1342).

### 3.4. Cytotoxic Evaluation against the Vero Cell Line

Cytotoxicity test of selected analogues was performed against the African green monkey kidney fibroblast (Vero cells) obtained from Bioassay laboratory, BIOTEC, NSTDA, Thailand. The cells of less than 20 passages were maintained continuously in a MEM/EBSS medium supplemented with 10% heated fetal bovine serum (GE Healthcare, PAA Laboratories GmbH, Pasching, Aurstria), 2.2 g/L sodium bicarbonate (Emsure, ACS, Reag. Ph Eur, Darmstadt, Germany), and 1% sodium pyruvate (Sigma). Cytotoxicity was determined by sulforhodamine B assay [45,46]. Briefly, 1.9 × 10^4^ Vero cells were incubated with each compound at final concentrations of 0.0001–100 µM at 37 °C under 5% CO_2_. After incubation for 72 h, the cells were fixed with 10% trichloroacetic acid (Sigma) at 4 °C for 45 min, washed gently with tap-water and air-dried at room temperature for overnight. Then, the fixed protein was stained with 0.057% *w*/*v* of sulforhodamine B (Sigma). The excess dye was washed repeatedly with 1% *v*/*v* of acetic acid. The plates were allowed to air-dry at room temperature for overnight. Finally, the protein-bound dye in each well was dissolved with 50 µL of 10× Tris-based solution. The optical density (OD) was measured by a microplate reader at wavelength 510 nm. The EC_50_ value of each compound was determined from the dose-response curve. Data were shown as mean and standard error of the mean (S.E.M.) of three biological independent experiments.

### 3.5. Cytotoxic Evaluation against the Vero E6 Cell Line

The cytotoxicity of the active compound was tested with Vero E6 cell lines. Each cell line was seeded at 1 × 10^4^ cells per well into 96-well plates in growth medium and incubated overnight. The compounds were prepared at the concentrations of 2–500 µM in DMSO and added to the cells at the DMSO final concentrations of 0.1%. Cells were incubated for 48 h before analyzing the cell viability using CellTiter 96^®^ AQueous One Solution Cell Proliferation Assay kit (Promega, Madison, WI, USA) according to the manufacturer’s protocol. The plate was read at the A450 by VICTORTM X3 microplate reader (PerkinElmer, Waltham, MA, USA). The EC_50_ values of the cells were calculated from nonlinear regression analysis. The results were reported as means and standard error of mean (S.E.M.) from three biological independent experiments.

### 3.6. In Vitro Antitrypanosomal Assay

*T. brucei rhodesiense* (STB900) was maintained continuously in MEM/EBSS medium (pH 7.3) supplemented with 3 g/L sodium bicarbonate, 4.5 g/L glucose, 25 mM HEPES, 0.05 mM bathocuproinedisulfonic disodium salt, 1.5 mM L-cysteine, 1 mM hypoxanthine, 0.16 mM thymidine, 1 mM sodium pyruvate, 0.2 mM 2-mercaptoethanol, 1% MEM non-essential amino acid, and 15% *v*/*v* heated (37 °C) FBS in a 5% CO_2_ incubator [47]. To evaluate the antitrypanosomal activity, trypanosome cells were seeded at 2 × 10^4^ cells/well in 175 µL culture media/well and incubated with 25 µL of varying concentrations of each test compound in a 96-well plate under the culture conditions. The compounds were dissolved in DMSO, whereby the final concentration of DMSO in each well was 0.1%, which did not affect the viability of the parasite. Following 72 h incubation, 20 μL of resazurin was added to each well. The reaction was further incubated at 37 °C under 5% CO_2_ for 3 h to allow for the irreversible reduction of resazurin (violet color) to resorufin (pink color) by viable trypanosome cells. Fluorescence signals were measured by a spectrofluorometer using a 530-nm excitation wavelength and a 585-nm emission wavelength. The results were reported as the concentration of each compound that exhibited a half-maximal inhibitory concentration (IC_50_) according to the dose-response curve derived from the fluorescence signals for each concentration of the compounds. The result for each compound was normalized using the control media for the overall background subtraction (0%), and untreated *T. brucei rhodesiense* with 0.1% DMSO was used as the 100% control. 

### 3.7. In Vitro Antimalarial Assay

The 3D7 *P.*
*falciparum* parasite was maintained continuously in human O+ erythrocytes (4% hematocrit) at 37 °C under 3% CO_2_ in RPMI 1640 culture media (pH 7.4) supplemented with 25 mM HEPES, 2 g/L NaHCO_3_, 40 mg/mL gentamicin, 0.37 mM hypoxanthine, and 5 g/L Albumax I. In vitro antimalarial activity was determined by malaria SYBR Green I-based fluorescence assay. Briefly, 0.09 mL of cultured 1% ring-stage synchronized parasites were transferred to the individual wells of a standard 96-well microtiter plate, and in vitro culture was continued for 48 h, with 0.01 mL of test compound at different concentrations in each well. The compounds were first dissolved and diluted in DMSO at various concentrations (0.0001–100 μM) and then in culture media to 10× concentration before being added to each well. The final concentration of DMSO in each well was 0.1%, which did not affect the viability of the parasite. After 48 h, SYBR Green I was added to each well, and the fluorescence signals were measured by spectrofluorometer using a 485-nm wavelength and a 535-nm emission wavelength. The results were determined as the IC_50_ concentration for each compound according to the dose-response curve established from the fluorescence signals for each concentration of the compounds. The result of each compound was normalized using the control media for the overall background subtraction as 0% and untreated parasite with 0.1% DMSO as 100% for the control [48]. 

### 3.8. Ethics Statement

Human erythrocytes were obtained from healthy volunteers aged 21–50 following the Thai Red Cross Society’s National Blood Center protocol. All the volunteers completed and signed a consent form before donating blood. The consent form and blood collection protocol were approved by the BIOTEC Ethics Committee (NIRB-024-2561).

### 3.9. In Vitro Anti-SARS-CoV-2 Main Protease (3CL^Pro^) Assay

The SARS-CoV-2 3CL^Pro^ activity assay was performed exactly as described [32]. The 3CL^Pro^ was prepared according to gene expression in *Escherichia coli* [49]. Initial rates (RFU/s) were measured in the absence and presence of compounds at 100 μM. The inhibition results were displayed as relative percentages compared to the initial rate in the absence of inhibitor. The assay was performed with 0.2 μM 3CL^Pro^, with 25 μME (EDANS)TSAVLQSGFRK(DABCYL) as the fluorogenic substrate in a phosphate buffer solution consisting of 1 mM dithiothreitol and 2% DMSO. Each test was performed in triplicate using a microplate reader (H1; BioTek Synergy). The volume was fixed at 100 μL, and the excitation and emission wavelengths were 340 and 490 nm, respectively. Initial rates relative fluorescence unit per second (RFU/s) were measured in both the absence and presence of the compounds at 100 μM. The inhibition results were calculated as a relative percentage compared to the initial rate in the absence of the inhibitor. Next, the active compounds having inhibitory activity greater than 60% were further evaluated the effective half-maximal inhibitory concentrations (IC_50_) using similar protocol with the test compound at various serial concentrations (10–500 μM). Data were shown as mean and standard deviation (S.D.) of three biological independent experiments.

### 3.10. In Silico Study of Compounds toward SARS-CoV-2 3CL^Pro^

In this study, the dimeric crystal structure of SARS-CoV-2 3CL^Pro^ in complex with a non-covalent inhibitor (X77) obtained from the protein databank with PDB ID: 6W63 [50] was used for the docking study of piperine (**1**), piperic acid (**2**), *N*-aryl amide analogs of piperine (**3**–**5**), and rutin using AutoDock Vina 1.2.1 [51] according to the standard protocols used in our previous studies [52,53]. All the ligand structures constructed by GaussView6 software were optimized by the B3LYP/6-31G* level of theory using the Gaussian09 program. The electrostatic potential charges (ESP) of the optimized ligands retrieved from the same level of theory were converted to the restrained electrostatic potential (RESP) charges using the antechamber module in AMBER20 [54]. All the complexes were performed by classical MD simulation for 100 ns [36,55]. The displacement between the centers of mass of ligand and the catalytic dyad (H41 and C145) was plotted along with the simulation time using the CPPTRAJ module. Only the simulations of the systems with ligands tightly attached to the 3CL^Pro^ binding pocket were extended to 200 ns. Per-residue decomposition free energy calculations based on the MM/GBSA method [56] were applied to 5000 snapshots from the last 50 ns to investigate the key binding residues. The 2D ligand-protein interaction pattern of the representative structure obtained from the RMSD clustering was visualized by BIOVIA Discovery Studio Visualizer V21.1.0 [57]. The UCSF Chimera V1.15 [58] was used to visualize all the 3D structures. The pharmacological properties of the compounds were predicted by SwissADME [28].

## 4. Conclusions

Piperine (**1**) is the major phytochemical isolated from the dried seeds of black pepper *(Piper nigrum*). Saponification of **1** generally gave piperic acid (**2**). In this study, compounds **3**–**5** were semi-synthesized by using **2** as the precursor under a modified esterification protocol involving carbodiimide and hydroxybenzotriazole as the coupling reagents. The two new *N*-aryl amide analogs of piperine that are the 2,4-dimethoxy phenyl amide analog of piperine (**4**) and 2,5-dimethoxy phenyl amide analog of piperine (**5**) were prepared in acceptable yields. The series of compounds **1**–**5** were evaluated for their cytotoxicity against the Vero and Vero E6 cell lines. All compounds were non-toxic at the concentration of 100 μM against selected kidney epithelial cells. Next, piperine (**1**), piperic acid (**2**) and *N*-aryl amide analogs of piperine (**3**–**5**) were examined antitrypanosomal, antimalarial, and anti-SARS-CoV-2 3CL^Pro^ activities. Compound **3** was highly active toward antimalaria (IC_50_ = 25.82 ± 1.56 μM). Compounds **4** and **5** exhibited interesting antiparasitic activity. Compound **4** demonstrated antitrypanosomal activity against *T. brucei rhodesiense* (IC_50_ = 19.87 ± 2.28 μM) and antimalarial activity against 3D7 strain *P. falciparum* (IC_50_ = 29.41 ± 3.54 μM). Compound **5** demonstrated antitrypanosomal activity at IC_50_ 15.46 ± 3.09 μM, and antimalarial activity at IC_50_ 24.55 ± 1.91 μM. Therefore, these two compounds have an advanced antiparasitic activity compared to piperine, which displayed antitrypanosomal activity at IC_50_ 56.67 ± 0.98 μM and antimalarial activity at IC_50_ 61.24 ± 2.83 μM. Compounds **1** and **5** both inhibited the activity of anti-SARS-CoV-2 3CL^Pro^ by approximately 70% at 100 μM. However, analog **5** showed the inhibition of 3CL^Pro^ at the IC_50_ value of 106.9 ± 1.2 μM, which was threefold more potent than the inhibitory profile of rutin, a previously reported 3CL^Pro^ natural inhibitor. Molecular docking and MD simulation were then performed to predict the binding mode of the compounds **1**–**5**. Regarding drug-likeness, ADME parameters, and pharmacokinetic property calculations demonstrated that compounds **4** and **5** contained higher hydrogen H-bond acceptors along with improved lipophilicity and water solubility better than piperine. The docking study revealed that compounds **1**–**5** were able to sit in the 3CL^Pro^ active site (PDB ID: 6W63). MD simulations were performed to calculate the ligand-binding stability. Piperine (**1**) and its analog **5** having 2,5-dimethoxy phenyl amide motif were relatively stable at 100 ns with the different binding interactions. Interestingly, compound **5** included three structural parts, (i) 1,3-benzodioxole group, (ii) pentadienamide, and (iii) 2,5-dimethoxy phenyl group, which played key roles in the binding interaction at the 3CL^Pro^ active site corresponding to alkyl–π and H-bond interaction in the presence of the unique sulfur–π interaction. Thus, the 2,5-dimethoxy phenyl amide analog of piperine (**5**) demonstrated promising potential as an anti-infective agent, and it could be developed for the treatment of infectious diseases such as human African trypanosomiasis, malaria, and COVID-19. 

## Figures and Tables

**Figure 1 molecules-27-02841-f001:**
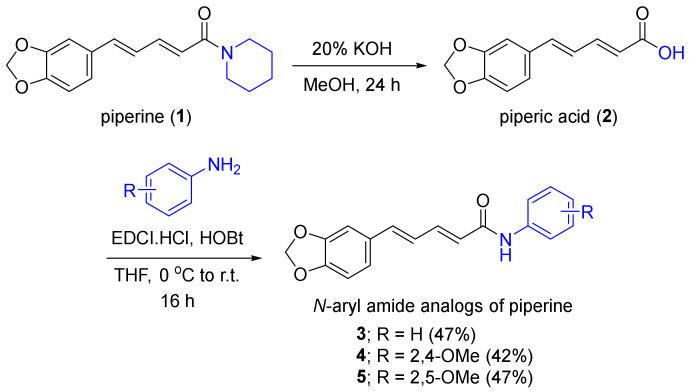
Synthesis of *N*-aryl amide analogs of piperine.

**Figure 2 molecules-27-02841-f002:**
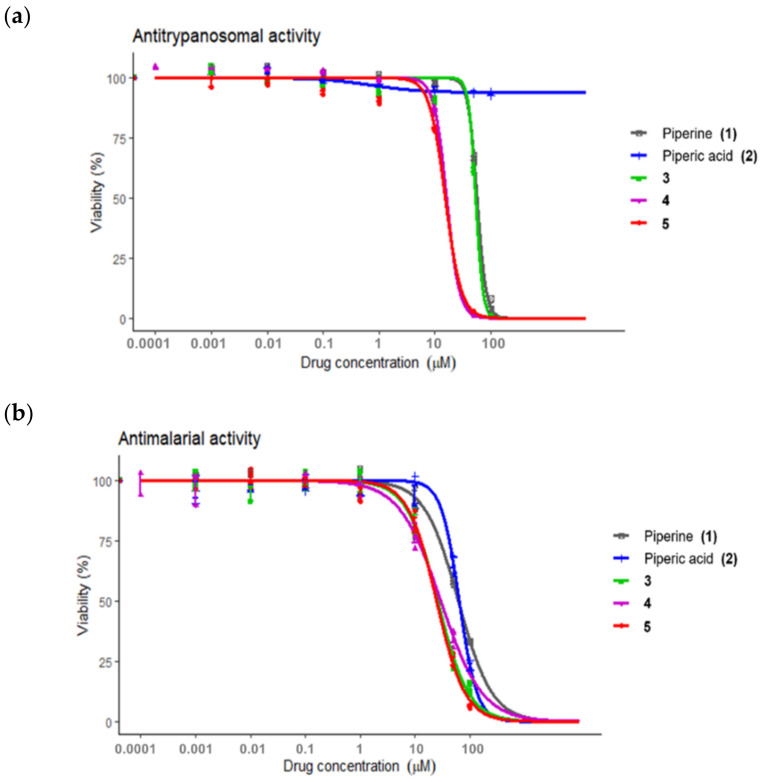
Antiparasitic activities of piperine (**1**), piperic acid (**2**), and *N*-aryl amide analogs of piperine (**3****–5**); (**a**) Antitrypanosomal activity toward the inhibition of *Trypanosoma brucei rhodesiense* and (**b**) antimalarial activity toward the inhibition of 3D7 *Plasmodium falciparum*.

**Figure 3 molecules-27-02841-f003:**
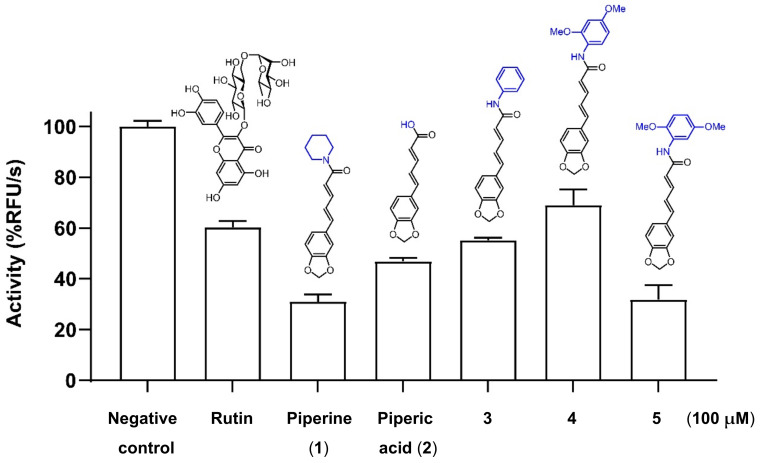
SARS-CoV-2 3C-like main protease inhibition of piperine (**1**), piperic acid (**2**), and *N*-aryl amide analogs of piperine **3****–5**. Compounds **1****–5** and rutin (positive control) were tested at 100 μM.

**Figure 4 molecules-27-02841-f004:**
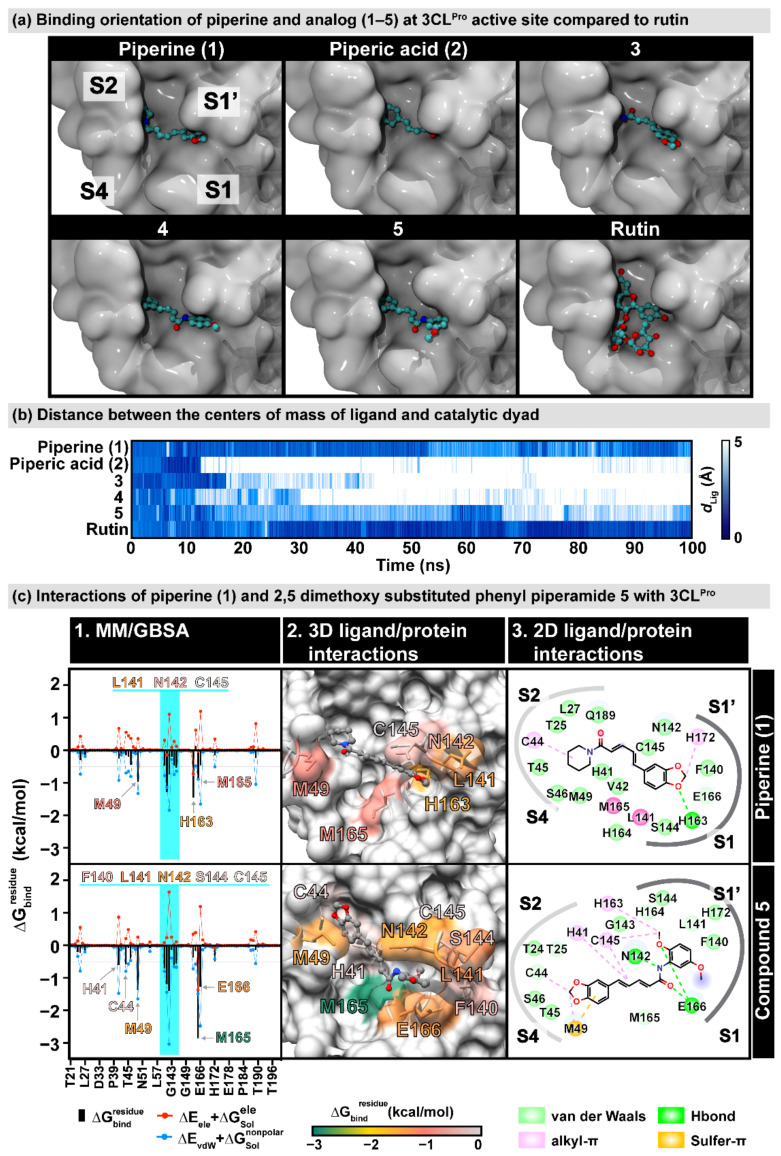
(**a**) Predicted binding orientation of piperine and analogs (**1**–**5**) at the active site of SARS-CoV-2 3C-like (3CL^Pro^) main protease obtained from molecular docking study compared to rutin [36]. (**b**) Distance between the center of the mass of the ligand and the catalytic dyad (H41 and C145) along with 100-ns molecular dynamics simulation of piperine and analogs (**1**–**5**). The colors are shaded from dark blue to white, indicating distances of 0 Å and 5 Å, respectively. (**c**) 3CL^pro^ interactions of piperine (**1**) and 2,5-dimethoxy substituted phenyl piperamide **5**. (**c**-**1**). MM/GBSA per-residue decomposition free energy ($\Delta G_{\mathrm{bind}}^{\mathrm{residue}}$) and its polar and nonpolar components of piperine (**1**) and 2,5-dimethoxy substituted phenyl piperamide (**5**). Residues with $\Delta G_{\mathrm{bind}}^{\mathrm{residue}}$ ≤ −0.5 kcal/mol are labeled, and the residues 140–145 in the oxyanion region are highlighted. (**c**-**2**). The 3D ligand-protein interactions are also shown according to $\Delta G_{\mathrm{bind}}^{\mathrm{residue}}$ values. (**c**-**3**). 2D ligand–protein interactions were drawn from the representative structure using BIOVIA Discovery Studio Visualizer V21.1.0.

**Table 1 molecules-27-02841-t001:** Cytotoxicity of piperine (**1**), piperic acid (**2**), and *N*-aryl amide analogs of piperine (**3**–**5**) against Vero and Vero E6 cell lines.

Compounds	EC_50_ ± S.E.M. (μM)	
	Vero	Vero E6
Piperine (**1**)	>100	131.67 ± 2.91
Piperic acid (**2**)	>100	>500
**3**	>100	>500
**4**	>100	>500
**5**	>100	>500

**Table 2 molecules-27-02841-t002:** Effective half-maximal inhibitory concentrations (IC_50_) of piperine (**1**), piperic acid (**2**), and *N*-aryl amide analogs of piperine (**3****–5**) against *Trypanosoma brucei rhodesiense* and *Plasmodium falciparum*.

Compounds	IC_50_ ± S.E.M. (μM)	
	Antitrypanosomal against *T. brucei rhodesiense*	Antimalarial against 3D7 *P. falciparum*
Piperine (**1**)	56.67 ± 0.98	61.24 ± 2.83
Piperic acid (**2**)	>100	63.54 ± 2.43
**3**	53.53 ± 1.77	25.82 ± 1.56
**4**	19.87 ± 2.28	29.41 ± 3.54
**5**	15.46 ± 3.09	24.55 ± 1.91
Pentamidine *^a^*	0.016 ± 0.002	n.d.
Pyrimethamine *^b^*	n.d. *^c^*	0.117 ± 0.017

*^a^*Positive control for antitrypanosomal; *^b^* Positive control for antimalaria; *^c^* n.d. refers to not determined.

**Table 3 molecules-27-02841-t003:** Inhibitory activity of SARS-CoV-2 3C-like main protease with piperine (**1**), piperic acid (**2**), and *N*-aryl amide analogs of piperine (**3**–**5**).

Compounds	Relative Protease Activity at 100 μM. (%RFU/s ^*a*^ ± S.D.)	IC50 ± S.D. (μM)
Rutin	60.3 ± 2.4	325.6 ± 12
Piperine (**1**)	31.0 ± 2.8	178.4 ± 1.2
Piperic acid (**2**)	46.9 ± 1.4	n.d. *^b^*
**3**	55.1 ± 1.1	n.d.
**4**	69.0 ± 6.1	n.d.
**5**	31.8 ± 5.7	106.9 ± 1.2

*^a^* RFU/s refers to relative fluorescence unit per second. *^b^* n.d. refers to not determined.

## Data Availability

Not applicable.

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
