# Peer review of "Semi-Synthesis of N-Aryl Amide Analogs of Piperine from Piper nigrum and Evaluation of Their Antitrypanosomal, Antimalarial, and Anti-SARS-CoV-2 Main Protease Activities"

_molecules, 2022, doi:10.3390/molecules27092841_

Round 1

Reviewer 1 Report

The authors present the preparation of three compounds derived of piperine using a synthetic strategy already described in several papers. Two of the compounds are new and the chemical and biological aspects described in the paper for the first time.  Additionally, they evaluate the antitrypanosomal, antimalarial and anti-SARS-COV2 activities. The manuscript is well written and structured, however, in my opinion the chemistry part is too poor, using a described strategy with a very few scope of only two compounds. Reinforcing that, the activity of both compounds was weak in all the applications tested, only showing activity using very high concentration of compounds (100 uM), without testing the cytotoxicity of that. 
For publish this work on molecules the authors should strongly improve the manuscript,  on the chemistry side by performing a larger scope of the methodology and on the biological side, testing the cytotoxicity  of the compounds in the range of concentrations used for observe activity. 

Author Response

Response to reviewer 1:

Response: Regarding reviewer comments, we sincerely appreciate your suggestions to improve the manuscript. In this work, the two new N-aryl piperamides were synthesized. Their biological activities against antitrypanosomal, antimalarial, and anti-SARS-Cov-2 were initially evaluated. Among the compounds in this work, 2,5-dimethoxy substituted phenyl piperamide 5 exhibited the most robust biological activities compared to the other compounds with the improved biological activities compared to piperine (1) and piperic acid (2), which are the natural compounds that originated for the black pepper seed.

We agreed to perform the cytotoxicity test of all compounds against Vero and Vero E6 cell lines (Table 1). We observed that piperine (1), piperic acid (2), and analogs 3-5 were non-toxic against Vero cells at the concentration of 100 mM. In addition, piperine (1) showed cytotoxicity at the half maximal effective concentration (EC50) of 131.67 ± 2.91 mM, while piperic acid (2) and analogs 3-5 were non-toxic to Vero E6 cell lines at the concentration of 500 mM. 

The results of cytotoxicity were described in lines 156-162 and Table 1 (lines 164-167). In addition, the cytotoxicity tests were explained in Materials and Methods at lines 443-472.

Anti-SARS-Cov-2 activities of piperine (1) and analog 5 were further evaluated to obtain the inhibition at the concentration of IC50 at 178.4±1.2 mM and 106.9±1.2 mM, respectively. Analog 5 improved anti-SARS-Cov-2 activity better than piperine (1), with threefold more potent than rutin (Table 2).

The results of the effective half-maximal inhibitory concentrations toward Anti-SARS-Cov-2 activities was described in in lines 238-241 and Table 2 (lines 257-260), along with Materials and Methods at lines 528-532.

Compound 5 exhibited non-toxicity against Vero and Vero E6 cell lines at 100 mM and 500 mM, respectively, which were in the concentrations used for observing antitrypanosomal, antimalarial, and anti-SARS-Cov-2 activities. Thus, 2,5-dimethoxy substituted phenyl piperamide 5 showed potential anti-infective agents with socio-economic advantages for treating human African trypanosomiasis, malaria, and COVID-19.

We revised the manuscript according to reviewer comments. All suggestions were addressed and showed in yellow highlighted contents.

Reviewer 2 Report

This is well designed project which addressed the pharmacological potential of piperine and its analogs. The manuscript provide ample of information. its a well written manuscript, however manuscript requires to improve the English language. eg

line 95; Rephrase the sentence 

line 103; "Piperine (1) was prepared from the dried black pepper seed" it should be isolated or extracted.

caption figure 1: "syntheses" change to synthesis 

figure 3. In the legends on x-axis, I suggest author to mention the concentration along with the name of compounds. Also mention working concentration of rutin in figure 3 and in text also.  

line 441: "The compounds were first dissolved and diluted in DMSO at 441
various concentrations ". mention the range of concentrations.

Author Response

Response to reviewer 2:

We revised the manuscript according to reviewer comments. All suggestions were addressed and showed in yellow highlighted contents.

Comment: Line 95; Rephrase the sentence 

Response: The sentence was rephases as follows.

Lines 103-104: We have prepared piperine (1) and semi-synthetic N-aryl amide analogs, followed by the evaluations of antitrypanosomal, antimalarial, and anti-SARS-CoV-2 main pro-tease 3CLPro activities. Furthermore, the binding interaction of the active piperine analogs with 3CLPro were demonstrated by docking and molecular dynamic (MD) simulation. The findings of this study would support the development of potential anti-infective agents with socio-economic advantages for the treatment of human African trypanosomiasis, malaria, and COVID-19.

Comment: Line 103; "Piperine (1) was prepared from the dried black pepper seed" it should be isolated or extracted. 

Response: The sentence was revised as follows.

Lines 109: Piperine (1) was isolated from the dried black pepper seed

Comment: Caption figure 1: "syntheses" change to synthesis 

Response: The caption of Figure 1 was revised as follows.

Lines 135:Figure 1 Synthesis of N-aryl amide analogs of piperine.

Comment: Figure 3. In the legends on x-axis, I suggest author to mention the concentration along with the name of compounds. Also mention working concentration of rutin in figure 3 and in text also. 

Response: Compounds 1-5 and rutin (positive control) were tested at 100 μM.

Line 234: We described concentration of rutin at 100 μM.

Line 235: We described concentration of compounds 1-5 at 100 μM.

Line 251: In the legend on x-axis, we added compound’s name and concentration at 100 μM.

Line 253: In the caption of Figure 3, we described concentration of rutin and compounds 1-5 at 100 μM.

Comment: Line 441: "The compounds were first dissolved and diluted in DMSO at various concentrations ". mention the range of concentrations.

Response: We added the range of concentration as follows.

Line 502: The compounds were first dissolved and diluted in DMSO at various concentrations (0.0001-100 mM)

Round 2

Reviewer 1 Report

The authors did the corrections of the reviews and performed additional citotoxicity studies, showing that the compounds are not toxic is the high concentrations used. However, only with a scope of two  compounds I do not think that is a paper for a journal with the impact factor of the Molecules. The authors should try the submission for a lower impact journal or really increase the scope on the synthetic part.